

# A study of institutional spending on open access publication fees in Germany

Najko Jahn[1] and Marco Tullney[2]

[1] Bielefeld University Library, Bielefeld University, Bielefeld, Germany
[2] German National Library of Science and Technology (TIB), Hannover, Germany

## ABSTRACT

Publication fees as a revenue source for open access publishing hold a prominent place on the agendas of researchers, policy makers, and academic publishers. This study contributes to the evolving empirical basis for funding these charges and examines how much German universities and research organisations spent on open access publication fees. Using self-reported cost data from the Open APC initiative, the analysis focused on the amount that was being spent on publication fees, and compared these expenditure with data from related Austrian (FWF) and UK (Wellcome Trust, Jisc) initiatives, in terms of both size and the proportion of articles being published in fully and hybrid open access journals. We also investigated how thoroughly self-reported articles were indexed in Crossref, a DOI minting agency for scholarly literature, and analysed how the institutional spending was distributed across publishers and journal titles. According to self-reported data from 30 German universities and research organisations between 2005 and 2015, expenditures on open access publication fees increased over the years in Germany and amounted to € 9,627,537 for 7,417 open access journal articles. The average payment was € 1,298, and the median was € 1,231. A total of 94% of the total article volume included in the study was supported in accordance with the price cap of € 2,000, a limit imposed by the Deutsche Forschungsgemeinschaft (DFG) as part of its funding activities for open access funding at German universities. Expenditures varied considerably at the institutional level. There were also differences in how much the institutions spent per journal and publisher. These differences reflect, at least in part, the varying pricing schemes in place including discounted publication fees. With an indexing coverage of 99%, Crossref thoroughly indexed the open access journals articles included in the study. A comparison with the related openly available cost data from Austria and the UK revealed that German universities and research organisations primarily funded articles in fully open access journals. By contrast, articles in hybrid journal accounted for the largest share of spending according to the Austrian and UK data. Fees paid for hybrid journals were on average more expensive than those paid for fully open access journals.

Corresponding author
Najko Jahn,
najko.jahn@uni-bielefeld.de

## INTRODUCTION

### General Background

The rise of open access journals has been matched by the increasing relevance of publication fees in academic publishing (*Davis & Walters, 2011*; *Laakso & Björk, 2012*; *Pinfield, 2015*). To cover these fees, also referred to as article-processing charges (APCs), authors tend to make use of funding that grant agencies or academic institutions provide (*Suber, 2012*). However, the question of how and to what extent these research support activities are effective in terms of the number of supported articles and their associated costs remains under debate.

The study of institutional spending on open access journal articles has been limited for several reasons. The first is that payment of these charges is fragmented across the budgets of grant agencies, research institutions, and libraries, or is covered by personal budgets. A comprehensive 2010 survey asked 9,645 authors from various disciplines how they financed publication fees, and it revealed that the majority of the respondents had access to research funding or institutional support to cover these charges. By contrast, 12% paid publication fees individually (*Dallmeier-Tiessen et al., 2011*). These results are consistent with similar findings from other studies: Previous studies also found that funding sources exist primarily in higher-income countries, mainly to support research articles in the biological and physical sciences (*Solomon & Björk, 2011*). Personal budgets, however, are likely used to cover lower publication fees (*Björk, 2015*; *Solomon & Björk, 2011*).

Another key problem in this regard is that funding for open access journals using publication fees lacks transparency because the parties involved—authors, universities, funders, and publishers—do not release information about who pays for what or the costs of publishing (*Björk & Solomon, 2014*); this situation is similar to the lack of transparency regarding journal subscriptions (*Lawson & Meghreblian, 2015*). To date, empirical studies examining publication fees have obtained price estimates by surveying authors (*Dallmeier-Tiessen et al., 2011*) or obtained them from journal websites. Using the latter method, two studies investigating journals across a broad range of disciplines calculated similar price averages that ranged between $904 (*Solomon & Björk, 2012*) and $923 (*Walters & Linvill, 2011*), as well as considerable price variation across journals and publishers. Accordingly, *Solomon & Björk (2012)* suggested using publication fees to cluster fully open access journal into several groups. In descending order, these are high-impact journals, followed by biomedicine journals from commercial publishers, large multi-disciplinary journals, and mid-price journals from commercial publishers covering a large spectrum of disciplines. Lower-priced journals are those published by academic societies and by publishers from low-income countries.

Nevertheless, it remains unclear which factors contribute to pricing in academic publishing. Generally, these might include article processing, impact, rejection rates, management and investment, and profit margins (*Noorden, 2013*). While fixed prices for individual articles are common, agreements between publishers and institutions can lead to discounts, and publishers sometimes waive publication fees for authors from low-income

countries (*Björk & Solomon, 2012*; *Lawson, 2015c*). Other factors leading to variable pricing schemes include submission or page charges (*Björk & Solomon, 2012*).

Hybrid journals substantially add to the complexity of open access funding (*Björk & Solomon, 2014*; *Kingsley, 2014*; *Pinfield, Salter & Bath, 2015*). These journals, which allow articles to be published immediately as open access after a charge is paid, rely on both subscriptions and publication fees as revenue sources. Although the uptake of open access through hybrid journals was described as lower and more expensive than that of fully open access journals (*Björk & Solomon, 2014*; *Solomon & Björk, 2012*), this model has gained increasing attention because of recent open access science policies, notably from the UK (*Pinfield, 2015*).

To address the problems of fragmented spending on publication fees and the lack of transparency about what is being paid, some European research funders and research-performing institutions have recently begun to disclose their expenditures for publication as open data. To the best of our knowledge, the first research funders to provide such data were the Wellcome Trust (*Kiley, 2014*) and the Austrian Science Fund FWF (*Reckling & Kenzian, 2014*). The not-for-profit company Jisc followed this example by collecting data from UK universities (*Lawson, 2015b*). Disclosed as publicly available spreadsheets, these data-sets self-report expenditures along with bibliographic information, including title, journal and publisher, and a persistent identifier for the publisher's version. Curatorial efforts focused on the disambiguation of publisher and journal titles as well as on detecting duplicates and persistent identifiers for the full text including the Digital Object Identifier (DOI) (*Neylon, 2014*; *Woodward & Henderson, 2014*). A preliminary version of Jisc's cost data was examined by *Pinfield, Salter & Bath (2015)*. Although the average spending on publication fees remained stable across universities, the authors found large price variations, as well as a varying number of articles supported by UK universities between 2007 and 2014, findings which confirm earlier studies that collected price information from journal websites (*Solomon & Björk, 2012*).

## Central funding for publication fees in Germany

This paper focuses on how much German universities and research organisations spend on open access publication fees. In Germany, the Deutsche Forschungsgemeinschaft (DFG), the largest German research funder, has strongly influenced how universities manage institutional support for these charges. Before the DFG started to support centrally funded publication fees through its "Open-Access Publishing" programme in 2011, only a few central funds existed (*Eppelin et al., 2012*). This is similar to the situation described in Canada (*Hampson, 2014*) and the UK (*Pinfield & Middleton, 2012*). The DFG has enforced a set of criteria with which grantees have to comply and which has resulted in similar policies regarding support for publication fees across German universities (*Fournier & Weihberg, 2013*). These criteria exclude the sponsorship of articles in hybrid journals and the funding of articles for which the publication fee exceeds € 2,000.[1] Grantees agree not only to pay for APCs, but also to find ways to improve the handling of those financial transactions. These ways include central invoicing schemes and memberships that are

[1]Guidelines for the funding program can be found here: http://www.dfg.de/formulare/12_20/.

agreed upon by university libraries and publishers and that often lead to a discount on publication fees (*Fournier & Weihberg, 2013*).

Non-university research organisations, i.e., institutes organised in the Fraunhofer-Gesellschaft, Helmholtz-Gemeinschaft, Leibniz-Gemeinschaft, and Max-Planck-Gesellschaft, are not eligible for this DFG funding programme. However, in response, some organisations have adopted similar processes to support authors. The Max-Planck-Gesellschaft operates its long-standing open access activities, including handling spending and publisher agreements centrally, through the Max Planck Digital Library (MPDL) (*Schimmer, Geschuhn & Palzenberger, 2013*; *Sikora & Geschuhn, 2015*), while the Leibniz-Gemeinschaft set up a dedicated open access fund in 2016.

The evolving institutional support for covering open access publication fees has led to calls for a unified approach towards open access funding in Germany. The MPDL called for re-allocating subscriptions in favour of open access journals in 2015 (*Schimmer, Geschuhn & Vogler, 2015*). At the same time, the Allianz der Wissenschaftsorganisationen, a science policy board representing all major research organisations in Germany, marked price transparency as one way to sustain an "adequate open access publication system" (*Bruch et al., 2015*). Reflecting Austrian and UK initiatives to share institutional spending on open access publication fees as open data, as well as professional discussions on open access publishing, Bielefeld University Library began to openly share its payment of publication fees in May 2014. After engaging with the working group "Electronic Publishing" of the Deutsche Initiative für Netzwerkinformation (DINI), other German institutions joined under the umbrella of the Open APC initiative soon after (*Apel et al., 2014–2016*).

### Research question

The aim of this study was to examine how much German universities and research organisations spent on open access publication fees. Using self-reported cost data from the Open APC initiative, the analysis focused on the amount that was being spent on publication fees, and compared these expenditure with data from related Austrian and UK initiatives in terms of both size and the proportion of articles being published in fully and hybrid open access journals. We also investigated how thoroughly self-reported articles were indexed in Crossref, a DOI minting agency for scholarly literature, and analysed how the institutional spending was distributed across publishers and journal titles.

## METHODS AND MATERIALS

We analysed self-reported cost data released by the Open APC initiative on May 13, 2016, (https://github.com/OpenAPC/openapc-de/releases/tag/v2.4.3) to assess institutional spending on open access publication fees in Germany. In addition to administrative data about the amount paid per article, including value-added tax, the reporting institution, and the year of invoicing, we used information about whether an article was published in a fully open access journal or in a hybrid journal, as well as the DOI reported in the data-set.

We obtained bibliographic metadata for each article from Crossref on May 19, 2016, on the basis of the reported DOIs. Although the Open APC initiative gathered metadata

representing publishers and journals from Crossref as well, this information was retrieved at the time when the participating institutions submitted their expenditure. To be transparent over time, the Open APC initiative kept track of the date when these data-sets were submitted with Git, a version control system that is increasingly used for enabling reproducible research (*Ram, 2013*), and made this information available via GitHub. However, during these data collection activities, Crossref regularly updated the metadata to represent ongoing mergers of publishing houses. A prominent example in this regard was the merger of the two large publishing houses Springer Business + Media and Nature Publishing Group announced on May 6, 2015, which operated as Springer Nature at the time of our study. To reflect these dynamics in academic publishing, we decided to retrieve updated bibliographic metadata from Crossref and to merge these records with the administrative information rather than re-using the historical publisher and journal information contained in the Open APC data-set.

We used the R package rcrossref (*Chamberlain et al., 2016*), developed and maintained by the rOpenSci initiative (https://ropensci.org/) to access Crossref's REST API (https://github.com/CrossRef/rest-api-doc/blob/master/rest_api.md). We requested the XML-based format `application/vnd.crossref.unixsd+xml` in which full and abbreviated journal titles as well as the ISSN media types (the International Standard Serial Number used to identify journals) were distinguished. This source also contained normalised publisher information, thus avoiding confusion regarding the naming of publishing houses that other studies faced when working with self-reported data (*Woodward & Henderson, 2014*). In cases where no bibliographic information could be obtained, we used the Open APC values. Because Crossref is not the only registration agency for DOIs –the agencies DataCite, Medra, and others also mint DOIs for scholarly works –we also identified the DOI agency for each article with the help of the rcrossref client.

Data collection also involved obtaining cost data from related open data initiatives. To compare self-reported spending on open access journal articles by German universities and research organisations with that of other initiatives, we consulted the openly available data-sets from the Austrian Science Fund (FWF) (*Reckling & Rieck, 2015*; *Rieck et al., 2016*), Jisc (*Lawson, 2015a*; *Lawson, 2016*) and the Wellcome Trust (*Kiley, 2015*; *Kiley, 2016*). For analysis, we obtained the overall publication fee spending on both fully and hybrid open access journals. In the case of FWF, we gathered the cost information from the accompanying spending reports. We used the spreadsheet data to calculate Wellcome Trust's and Jisc's spending, and converted the prices from GBP to Euro in accordance with the average foreign exchange reference rates provided by the European Central Bank. Our comparison of the open data initiatives focused on the last two years: 2014 and 2015. Because Wellcome Trust's spending was reported for the fiscal periods 2013–2014 and 2014–2015, we referred to the average exchange rates of the full two-year period as we could not determine the actual invoicing dates from the data. We excluded articles from the analysis with missing information about the cost or the journal type. In the case of Jisc's 2014 data (*Lawson, 2015a*), for instance, we excluded spending on 2,812 publications that amounted to € 4,861,772 from the analysis because no publication type was given in the data-set.

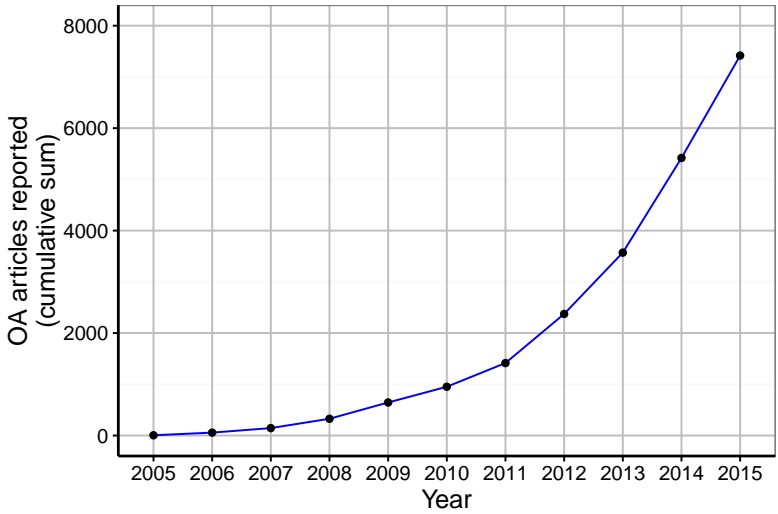

**Figure 1** Growth of the Open APC Initiative.

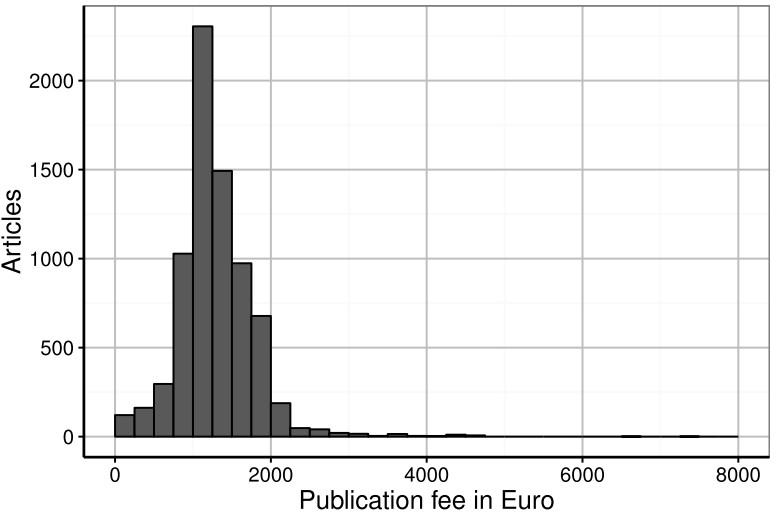

**Figure 2** Institutional spending on publication fees by German research organisations per article (in €).

Data curation activities of the Open APC initiative and those of the other initiatives differed in some respects. For instance, the DOI was a mandatory element in the Open APC data template that the participating institutions were required to report, whereas in the case of the Wellcome Trust spending data DOIs were additionally identified by automated compliance checks. Our first screening of the data-sets revealed that some articles published in Crossref-indexed journals lacked a DOI. Because of these different methods to curate the cost data and because our main focus was institutional funding for publication fees in Germany, we decided to compare only German spending with that reported by other initiatives. We did not, therefore, analyse the distribution of spending over publishers and journal titles or the indexing coverage in Crossref for the Austrian and UK spending data.

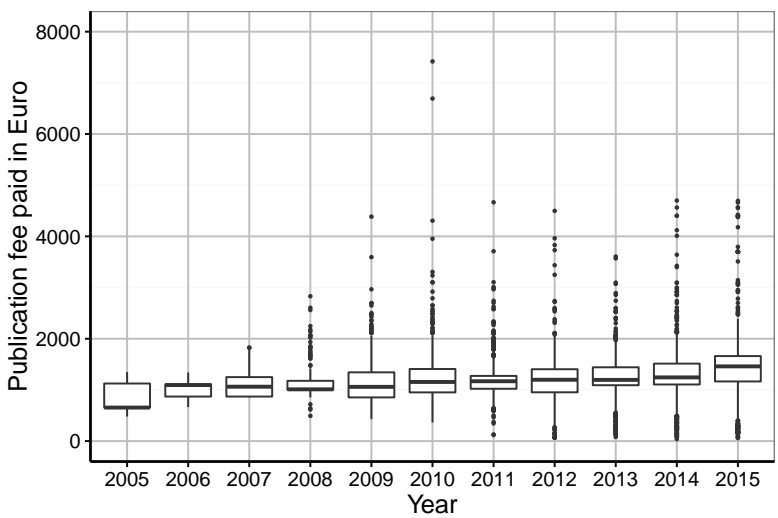

**Figure 3** Institutional spending on publication fees by German research organisations per year (in €).

# RESULTS

## Cost Data

After excluding payments for non-journal articles as well as articles invoiced in 2016, we retrieved information on 7,417 open access journal articles that 30 German universities and research institutions supported financially between 2005 and 2015. As illustrated in Fig. 1, payments made for open access journal articles increased over the years. While one institution supported five articles in 2005, most institutions included in our study shared their expenditure from 2013 onwards. The best represented year was 2015, with 1,999 articles. However, at the time of analysis only 27 institutions had contributed cost information for 2015, suggesting a lag between the time that payments are made and expenditures are reported to the Open APC initiative.

The fees for all of the articles amounted to € 9,627,537, including value-added tax; the average payment was € 1,298 (median = € 1,231, SD = € 486). Figure 2 presents the distribution of institutional spending on publications. We observed that 6,996 (94%) of the publication fees were paid in accordance with the DFG price cap of € 2,000. Most payments for publications ranged from € 1,000 to € 1,250.

Figure 3 presents institutional spending per article and year. Large price variations can be observed. Publication fees that were paid by German universities and research organisations ranged from € 40 to € 7,419. However, the average price paid varied somewhat during the period from 2011 to 2015 (€ 1,239–€ 1,423).

The number of APC payments per institution varied considerably (see Table 1). With 2,856 reported articles, the Max Planck Society contributed 39% of the overall article volume. By contrast, we observed a lower number of supported open access journal articles for several universities that had only recently begun to set up centrally managed open access funds to cover publication fees.

**Table 1  Institutional spending on open access publications (in €).**

| Institution | Articles funded | Total | Mean | SD | Median | Min–Max |
|---|---|---|---|---|---|---|
| MPG | 2,856 | 3,661,120 | 1,282 | 464 | 1,168 | 69.12–7,418.88 |
| Goettingen U | 650 | 883,918 | 1,360 | 476 | 1,354 | 180–4,694.83 |
| KIT | 426 | 523,166 | 1,228 | 525 | 1,243 | 69–3,731.09 |
| Regensburg U | 399 | 503,205 | 1,261 | 503 | 1,207 | 77.35–4,403 |
| Muenchen LMU | 365 | 463,491 | 1,270 | 296 | 1,299 | 496–2,023 |
| TU Muenchen | 308 | 390,086 | 1,267 | 479 | 1,386 | 130.82–2,121.77 |
| Bielefeld U | 262 | 322,815 | 1,232 | 305 | 1,234 | 142–2,103 |
| Giessen U | 243 | 326,082 | 1,342 | 583 | 1,247 | 80.92–4,498.2 |
| Konstanz U | 221 | 302,659 | 1,369 | 404 | 1,380 | 40–2,071.51 |
| Heidelberg U | 215 | 308,348 | 1,434 | 377 | 1,500 | 59.5–2,042 |
| Wuerzburg U | 207 | 286,543 | 1,384 | 429 | 1,447 | 105.07–2,514.09 |
| Leipzig U | 173 | 243,873 | 1,410 | 331 | 1,471 | 340.74–2,055.15 |
| FZJ - ZB | 158 | 196,869 | 1,246 | 516 | 1,177 | 369.69–3,700 |
| TU Dresden | 130 | 175,723 | 1,352 | 416 | 1,415 | 200–2,193.17 |
| Duisburg–Essen U | 114 | 136,911 | 1,201 | 302 | 1,214 | 238–1,982 |
| FU Berlin | 106 | 142,671 | 1,346 | 466 | 1,292 | 219.84–2,000 |
| GFZ-Potsdam | 106 | 126,520 | 1,194 | 760 | 1,065 | 222.53–4,403 |
| Bayreuth U | 92 | 105,725 | 1,149 | 532 | 1,200 | 81.86–2,058.7 |
| Bochum U | 71 | 93,546 | 1,318 | 460 | 1,438 | 100–2,041.64 |
| Hannover U | 69 | 90,259 | 1,308 | 414 | 1,241 | 148.75–2,158.97 |
| MDC | 69 | 145,256 | 2,105 | 1,228 | 1,800 | 490.58–4,699.61 |
| TU Chemnitz | 36 | 37,826 | 1,051 | 703 | 1,142 | 77.81–2,122.81 |
| Kassel U | 35 | 35,550 | 1,016 | 475 | 1,142 | 150–1,861 |
| Hamburg TUHH | 24 | 32,789 | 1,366 | 499 | 1,466 | 300.05–2,027.31 |
| Potsdam U | 24 | 32,128 | 1,339 | 236 | 1,386 | 916.3–2,116.45 |
| Bamberg U | 22 | 23,663 | 1,076 | 563 | 1,009 | 90–2,010 |
| TU Ilmenau | 13 | 13,053 | 1,004 | 617 | 986 | 178.5–2,076.55 |
| Dortmund TU | 9 | 8,238 | 915 | 566 | 900 | 155.1–1,738.06 |
| TU Clausthal | 8 | 6,999 | 875 | 514 | 918 | 180.94–1,723.64 |
| INM - Leibniz-Institut für Neue Materialien | 6 | 8,505 | 1,418 | 751 | 1,492 | 236.75–2,453.99 |

## Comparison of related cost data-sets

Table 2 compares the Open APC spending data with that of the Austrian FWF, as well as with Jisc's and Wellcome Trust's expenditures. Prices were converted according to the average Euro exchange rate during the examined periods and were gathered for both fully open access journals and hybrid journals. The comparison revealed that the Open APC initiative lacked cost information about hybrid journals, whereas the related Austrian and UK open data initiatives reported a large share of spending on these journals between 2014 and 2015. Over the years 2005–2015, three out of 30 German universities and research institutions reported 60 hybrid journal articles to the Open APC initiative, representing 0.81% of all articles included in the data-set. In contrast, in terms of the number of supported articles and the amount spent on publication fees, the Open APC data-set

**Table 2  Comparison of cost data per period and journal type (in €).**

| Cost data-set | Journal Type | Articles funded | Total costs in € | Mean |
|---|---|---|---|---|
| **FWF** | | | | |
| **2014** | Fully OA | 247 | 316,765 | 1,282 |
| | Hybrid OA | 780 | 1,794,604 | 2,301 |
| **2015** | Fully OA | 288 | 418,408 | 1,453 |
| | Hybrid OA | 912 | 2,376,356 | 2,606 |
| **Jisc** | | | | |
| **2014** | Fully OA | 1,161 | 1,897,862 | 1,635 |
| | Hybrid OA | 2,938 | 5,409,623 | 1,841 |
| **2015** | Fully OA | 1,168 | 2,211,958 | 1,894 |
| | Hybrid OA | 2,944 | 6,977,753 | 2,370 |
| **Open APC** | | | | |
| **2014** | Fully OA | 1,832 | 2,353,665 | 1,285 |
| | Hybrid OA | 15 | 26,546 | 1,770 |
| **2015** | Fully OA | 1,991 | 2,820,445 | 1,417 |
| | Hybrid OA | 8 | 23,412 | 2,927 |
| **Wellcome Trust** | | | | |
| **2013–2014** | Fully OA | 607 | 911,302 | 1,501 |
| | Hybrid OA | 1,894 | 4,648,878 | 2,455 |
| **2014–2015** | Fully OA | 775 | 1,418,097 | 1,830 |
| | Hybrid OA | 2,065 | 5,690,178 | 2,756 |

provided more comprehensive price information for fully open access journals than did the Austrian and UK initiatives.

A comparison of average prices revealed that publishing in hybrid journals was, on average, more expensive than publishing in fully open access journals. Price differences between these two categories were also reported earlier, indicating that prices for fully open access journals were lower on average (*Pinfield, Salter & Bath, 2015*; *Solomon & Björk, 2012*). In 2014 and 2015, the mean price for fully open access journals calculated from all data-sets was below the DFG price cap of € 2,000.

## Crossref indexing

To identify publication fee spending at article level and to gather bibliographic metadata, DOIs were a mandatory part of the Open APC initiative's data collection activities. The participating institutions reported DOIs for 7,373 out of 7,417 articles. Using these DOIs, we retrieved additional metadata from Crossref for 7,346 publications, representing 99% of the total article volume. Articles for which no metadata could be obtained, were registered with the DOI agency DataCite (ten articles) or Medra (two articles). For eight articles, our parser could not gather the XML resource, although these publications were registered with Crossref at the time of our study. Seven DOIs reported to the Open APC initiative could not be resolved.

**Table 3  Publication fees paid per publisher (in €).**

| Publisher | Articles funded | Total | Mean | SD | Median | Min–Max |
|---|---|---|---|---|---|---|
| Springer Nature | 2,167 | 2,948,697 | 1,361 | 387 | 1,385 | 80.92–4,403 |
| Public Library of Science (PLoS) | 1,680 | 2,243,128 | 1,335 | 321 | 1,207 | 555.66–2,790.27 |
| Frontiers Media SA | 906 | 1,186,283 | 1,309 | 424 | 1,142 | 77.35–4,179 |
| Copernicus GmbH | 841 | 1,160,450 | 1,380 | 658 | 1,277 | 69.12–7,418.88 |
| IOP Publishing | 677 | 699,137 | 1,033 | 228 | 953 | 374.77–1,950 |
| MDPI AG | 208 | 236,729 | 1,138 | 453 | 1,177 | 154.43–2,054.68 |
| Hindawi Publishing Corporation | 120 | 125,495 | 1,046 | 538 | 947 | 174.99–2,225.22 |
| The Optical Society | 111 | 176,665 | 1,592 | 392 | 1,626 | 498.62–3,731.09 |
| Wiley-Blackwell | 78 | 126,148 | 1,617 | 467 | 1,601 | 490.58–3,065 |
| Oxford University Press (OUP) | 64 | 118,225 | 1,847 | 793 | 1,741 | 297.5–4,498.2 |
| Other | 565 | 606,578 | 1,074 | 840 | 922 | 40–4,699.61 |

## Cost data by publisher and journal

We used the DOI to automatically fetch publisher and journal names for each article from the Crossref REST API. Table 3 shows the top ten publishers in terms of the number of financially supported articles. These publishers represented 92% of all articles included in our data-set. In total, payments were made to 139 publishing houses. Comparing these data with data from the UK, we observed that a greater share of total spending was directed to some open access publishers. *Pinfield, Salter & Bath (2015)*, for instance, reported remarkably lower proportions for the open access publishers MPDI, Copernicus, and Hindawi.

Most of the publication fee spending in Germany was on articles published in Springer Nature journals, which likely reflects the results of mergers with the open access publisher BioMed Central in 2008 and between the well-established publishers Springer Science + Business Media and Nature Publishing Group in 2015. Using the Crossref-Member-ID instead of the publisher name, we were able to differentiate between journals formerly published by Springer Science + Business Media and Nature Publishing Group. In terms of articles, the majority of payments made were for publications in journals formerly associated with Springer Science + Business Media. Springer Science + Business Media journals accounted for 2,027 articles, representing 94% of the overall Springer Nature article volume recorded by the Open APC initiative and 92% of the amount that was spent. Median publication fee spending differed slightly between Springer Science + Business Media (€ 1,355 €) and Nature Publishing (€ 1,386). However, the price variation was higher for Nature Publishing journals (SD = € 848) than for the former Springer Science + Business Media titles (SD = € 313).

In contrast to Springer Nature, other well-established publishing houses such as Elsevier and Wiley-Blackwell ranked lower in our analysis.

**Table 4  Publication fees paid per journal (in €).**

| Journal | Articles funded | Total | Mean | SD | Median | Min–Max |
|---|---|---|---|---|---|---|
| PLOS ONE | 1,433 | 1,745,513 | 1,218 | 130 | 1,198 | 748.71–1,808.8 |
| New Journal of Physics | 673 | 693,322 | 1,030 | 225 | 953 | 374.77–1,856.4 |
| Atmospheric Chemistry and Physics Discussions | 281 | 437,903 | 1,558 | 776 | 1,403 | 233.86–7,418.88 |
| Frontiers in Psychology | 271 | 363,794 | 1,342 | 429 | 1,142 | 77.35–2,122.81 |
| BMC Genomics | 135 | 179,592 | 1,330 | 205 | 1,276 | 920–1,926 |
| Biogeosciences Discussions | 127 | 187,716 | 1,478 | 548 | 1,313 | 663.55–3,641.47 |
| BMC Bioinformatics | 113 | 142,680 | 1,263 | 217 | 1,244 | 655–1,661.24 |
| Frontiers in Plant Science | 107 | 126,763 | 1,185 | 408 | 1,106 | 551.04–2,380 |
| Atmospheric Measurement Techniques Discussions | 107 | 143,782 | 1,344 | 585 | 1,203 | 428.4–3,709.44 |
| Frontiers in Human Neuroscience | 106 | 140,065 | 1,321 | 415 | 1,106 | 575–2,000 |
| Other | 4,064 | 5,466,407 | 1,345 | 557 | 1,350 | 40–4,699.61 |

Prices also varied within journals. Table 4 illustrates the top ten out of 732 journals based on the number of supported articles. We normalised PLOS journal titles because the name change from "PLoS" to "PLOS" was only partially represented in the Crossref metadata at the time of our study. Articles published in the top ten journals represented 45% of the overall article volume. The multidisciplinary journal PLOS ONE ranked highest. In addition, the journals New Journal of Physics, Atmospheric Chemistry and Physics Discussions and Frontiers in Psychology, all of which publish contributions from all branches of their respective discipline, were also well represented in our study. In the case of Atmospheric Chemistry and Physics Discussions, the large price range can be explained by the fact that this journal charges per page and takes the submission's file format into consideration.

## DISCUSSION

In Germany, institutional spending on publication fees charged by open access journals has increased over the years. These findings are consistent with the general trend towards using publication fees as a revenue source for open access publishing (*Davis & Walters, 2011*; *Laakso & Björk, 2012*; *Pinfield, 2015*). They also demonstrate the growing trend among institutions in Germany to both encourage their researchers to publish in open access journals and to offer financial support (*Fournier & Weihberg, 2013*). Similar to the expenditures on publication fees at an institutional level in the UK (*Pinfield, Salter & Bath, 2015*), spending volume varies across German universities and research organisations. With a proportion of 39% of the total article volume, the Max Planck Society, a large non-university research organisation, supported the most open access journal publications included in our study. A possible explanation could be the centralised library support at the Max Planck Society, where the Max Planck Digital Library has managed open access agreements with publishers over the last decade on behalf of most Max Planck institutes

(*Schimmer, Geschuhn & Palzenberger, 2013*; *Sikora & Geschuhn, 2015*). This centralised approach presumably resulted not only in a large number of supported open access articles but also in central access to cost data provided by publishers, as well as in posessing the advanced capabilities and skills needed to report these expenditures on a regular basis. Many universities and research organisations, by contrast, disclosed a remarkably lower number of supported articles.

Re-using DOIs to gather bibliographic metadata from Crossref is a promising approach to addressing data curation issues raised by UK initiatives (*Neylon, 2014*; *Woodward & Henderson, 2014*). In our study, Crossref thoroughly indexed open access journal articles disclosed in the Open APC data-set, providing information about publisher and journal titles for 99% of all articles included in the Open APC data-set. Making use of metadata from Crossref, therefore, reduces the need for extensive validation of bibliographic records as long as the DOIs are made available in the cost data. Beyond identifying publishers and journals, mandatory reporting of DOIs in the spending data can also increase the use of such data to study other aspects of APC-based open access journals. For instance, impact analyses in the field of altmetrics make heavy use of DOIs as well (*Haustein, 2016*). Furthermore, drawing on Crossref has the potential to increase the comparability of cost data to prepare for future negotiations with publishers regarding open access agreements because Crossref's metadata represent current developments in academic publishing in terms of ongoing mergers of publishing houses. In addition to these benefits, future comparative studies of publication fee spending using data at the article level can also benefit from such an approach.

This study is limited in some respects. First, we cannot assess whether publishers and journals granted publication fee discounts. The Open APC initiative uses a minimal data scheme to encourage self-reporting, and it therefore does not track this type of information. However, large price variations suggest that different pricing levels and pricing schemes are in place, as previously observed (*Pinfield, Salter & Bath, 2015*; *Solomon & Björk, 2012*). Adding to this complexity, it is likely that some institutions paid only part of the publication fee. Consider, for instance, the journal Nature Communications, a journal that can be categorised as a pricy, high-impact journal according to *Solomon & Björk (2012)*: Charges reported to the Open APC initiative ranged between € 2,000, the DFG price cap, and € 4,403. Although making such payments from several budgets is a proposed strategy to sustain publication funds at German universities (*Fournier & Weihberg, 2013*), this splitting of payments was not made transparent in the Open APC data, leading to a possibly flawed representation of publication fee spending in Germany. In another case, one university included its contributions to the SCOAP[3] consortia and presumably divided the sum by the number of articles published by their authors in SCOAP[3]-covered journals.[2] This approach is arbitrary, because averages for an institution can be determined only after the end of a full 3-year funding cycle. Other factors affecting price variations are exchange rates and different tax rates for some organisations in Germany. For instance, the Max Planck Society has a limited input tax reduction. The refund of input value-added tax for publication fees is 20%. To increase the transparency of publication fee spending, *Pinfield, Salter & Bath (2015)* suggested disclosing tax rates and payment currencies in future cost

---

[2]SCOAP[3], the Sponsoring Consortium for Open Access Publishing in Particle Physics, is a unique approach to convert former subscription journals in high-energy physics to open access journals under a CC BY license, see https://scoap3.org/what-is-scoap3/. The consortium, led by CERN, pays publishers centrally, based on previously agreed APCs and an overall price cap, and retrieves its funds from organisations and countries based on their share in the articles published in the covered journals. German universities participate through an initiative led by the German National Library of Science and Technology that received additional funding from DFG.

data-sets. Likewise, the Open APC data-set does not track funding sources; thus, we could not determine which funders co-financed publication fees.

It must also be noted that reporting to the Open APC initiative is voluntary. Therefore, not all institutions in Germany that provide central funding for publication fees contribute cost data to this initiative. According to a qualitative survey that asked why German institutions are reluctant to share their cost data through the Open APC initiative, one institution feared that an increase in transparency would allow publishers to adjust prices in their favour. Others noted that the workload to produce such a data-set could be too extensive (*Deppe, 2015*). As no reliable registry of institutional open access funds or related support structures in Germany exists, we cannot assess the number of non-participants in Germany.

Our analysis of how institutional spending for open access articles was distributed over publishers and journals indicated that open access publishing is heterogeneous and concentrated at the same time. While we were able to identify 139 individual publishing houses that were supported by the German universities and research organisations, the distribution is highly skewed. Ten publishers collected 92% of open access publication fee spending, which is consistent with an observed high concentration of a few publishers in current academic publishing (*Larivière, Haustein & Mongeon, 2015*). However, our study could not confirm that publications in open access journals owned by traditional publishing houses accounted for most of the spending on publication fees as observed by *Pinfield, Salter & Bath (2015)*. Rather, open access publishers such as Public Library of Science (PLoS), Copernicus, or MPDI ranked higher in our study than they did in the analyses of cost data in the UK.

One possible explanation for why traditional publishers are less well represented in our study is the lack of cost information about hybrid open access journals. In fact, 99% of all articles that German universities and research organisations disclosed were published in fully open access journals. This result presumably reflects the DFG funding programme that excludes paying for open access articles published in hybrid journals. However, while reviewing self-reported cost data from Austria and the UK, where hybrid open access journals are generally supported, we observed a much higher share of payments for articles in hybrid open access journals. Because publication fee spending is fragmented and often lacks transparency, it remains open to speculation whether authors affiliated with German universities and research organisations avoid opting for open access when publishing in hybrid journals or whether they simply use other budgets that are not covered by the Open APC initiative.

## CONCLUSION

Our study revealed the size and extent of spending on open access journals using publication fees in Germany. According to self-reported cost data from the Open APC initiative, this type of support from German universities and research institutions has grown over the years. Comparing these expenditure with those from Austria and the UK, German open access funding is focused primarily on fully open access journals, raising important questions

about hybrid open access journals as a publication venue. Given our findings and the general discussion about funding policies addressing hybrid open access journals, questions about whether and to what extent science policies and the availability of institutional support influence how researchers publish are of particular concern.

Using self-reported data and gathering publisher and journal information from Crossref, our study extends methods and improves data collection activities for researchers and practitioners alike, as well as contributing to a better understanding of the factors affecting the analysis of publication fees in open access publishing. In this regard, our research highlights large variations in the distribution of spending that need to be taken into consideration when studying payments on publications at the institutional level. We have also confirmed the findings of other studies that showed large price variations across publishers and open access journals. This complex situation of fee-based open access publishing needs to be better understood by both researchers and practitioners.

## ACKNOWLEDGEMENTS

We thank Andrea Hacker and Ada-Charlotte Regelmann for their valuable comments on the first draft of this paper. We also thank Christoph Broschinski, Vitali Peil, Dirk Pieper, the members of the DINI working group "Electronic publishing," and all data contributors (https://github.com/OpenAPC/openapc-de#contributors) of the Open APC initiative.

### Funding

NJ acknowledges support from the Deutsche Forschungsgemeinschaft (DFG) under the project "INTACT—Transparente Infrastruktur für Open-Access-Publikationsgebühren." Article Processing Charge funded by the Deutsche Forschungsgemeinschaft and the Open Access Publication Fund of Bielefeld University. The funders had no role in study design, data collection and analysis, decision to publish, or preparation of the manuscript.

### Grant Disclosures

The following grant information was disclosed by the authors:
Deutsche Forschungsgemeinschaft (DFG).

### Competing Interests

The authors are involved in the Open APC initiative. The authors declare there are no competing interests.

### Author Contributions

- Najko Jahn conceived and designed the experiments, performed the experiments, analyzed the data, contributed reagents/materials/analysis tools, wrote the paper, prepared figures and/or tables, reviewed drafts of the paper.
- Marco Tullney conceived and designed the experiments, wrote the paper, reviewed drafts of the paper.

**Data Deposition**

Code repository hosted on GitHub:

https://github.com/njahn82/paper_openapc.

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
