# Peer review of "A study of institutional spending on open access publication fees in Germany"

_PeerJ, doi:10.7717/peerj.2323_

## Round 0.1 · original submission · Major Revisions

The paper is, in general, clearly written and appears to adhere to PeerJ policies concerning data access and availability as well as for software access for the API. However, the reviewers in general felt that there was not a clear linkage to the original research question (lines 112-119) for the paper, this is exclusive of the overall goals for APC. In essence there seems to be enough here for two papers, a technical paper on the workflow, collection, and digital preservation of APC data and a paper on the data results that can be derived from the current Results section of the paper. Question, Is this a technical paper to describe data collection workflows and methdology for APC or is there a research question that correlates to the current results (lines 224-279). In the Methods section as currently stated, the paper is about the APC workflow (lines 114-116). However, if the Methods section was retooled to look at the key questions answered by the Results section, we feel that it could be accepted as a PeerJ article under the current editorial guidelines. I feel this would be a major revision to the paper as it is currently constructed.

·

Basic reporting

This paper presents an effort to track how much money universities in Germany spend on open access publishing fees. The background is well-researched and acknowledges both the need for transparency in the funding of open access publications and existing efforts to gather and make available this kind of data. The figures and tables are helpful and the raw data is available in GitHub.

The paper is hindered by frequent grammatical and spelling errors, and the overall flow is hard to follow. The "Conclusions" section from the standard template is missing. Considerable editing in these areas would be required before publishing this paper.

Experimental design

The experimental design is well thought out and follows best practices for open data projects. The research question is clearly articulated, and the methods are described in detail.

Validity of the findings

Conclusions of the study are not stated. Questions that could be addressed in a revision of this paper include:

-How does the data from this study compare to that of the other efforts cited in the introduction (Wellcome Trust, Jisc, etc.)?
-What benefits/drawbacks were found using a bottom-up approach rather than a top-down one?
-How could this model be adapted or extended beyond Germany?
-What, if anything, does the data indicate about trends in open access publishing?

Reviewer 2 ·

Basic reporting

The paper is, in general, clearly written and appears to adhere to PeerJ policies.

Experimental design

I question whether this paper is in scope for PeerJ. It is a descriptive case report, without a clearly defined research question. This would be acceptable for publication in my field (LIS), but does not seem to meet PeerJ's guidelines.

Validity of the findings

Prior to the Results section, the paper reads as if Open APC is a technical project to aggregate cost data (lines 97-103), and describes the intent of the paper as to describe the technical workflow of the initiative (lines 104-106). The results section, however, focuses largely on presenting the data that were aggregated. While one would expect such data to be collected as a result of the workflow, I also would have expected a more technical presentation of the work (e.g., data on effectiveness of the workflow) given the earlier definition of the scope of the paper. The results align with the stated goals of the Open APC initiative, but in my opinion do not align with the stated goals of the paper.

With that said, I think the descriptive data presented are interesting and I'd recommend revising the introduction and methods to be more inclusive of the analysis and re-use of the data (and not in the technical sense already in the paper) as part of the Open APC initiative so the results section is more consonant with the stated intent of the paper. Alternately, the results section should be revised to better reflect the technical focus of the rest of the paper.

Additional comments

There is good content in the draft, but it could be improved by better aligning the results section with the rest of the paper.

·

Basic reporting

This article is a thorough, yet concise, overview of the Open APC initiative. The introduction provides a solid background into open access publishing models and the reader does not need any previous exposure to OA in order to understand the implications of Open APC. The authors have succeeded in not only explaining the initiative, but also offering insight into transparency in research. The explanation and implementation of tools like GitHub and CrossRef make this piece relevant for anyone engaged in digital scholarship.

Experimental design

The methods are described in great detail and this investigation could easily be carried out at other organizations or institutions.

Validity of the findings

This piece would benefit greatly from speculation. The details of the Open APC are thoughtfully laid out, but there is little to no discussion of what it might mean. The authors clearly state that further analysis of the data is needed in the future, but for now a discussion of potential implications would make this article more meaningful. A simple paragraph or two in the discussion portion would really strengthen this investigation.

---

## Round 0.2 · accepted · Accept

Thanks again for considering our recommendations. I believe this is a much tighter paper that does great job of explaining your research question, methodologies for data collection and in offering users a transparent way to replicate this collection methodology for the future. Your rebuttal was excellent and enabled us to review the revision in a short period of time.

·

Basic reporting

The scope of this paper has been heavily refined from the previous version so that its main focus is on the funding of APCs in Germany, and all discussion of the technical workflow of the Open APC Initiative has been removed. These changes have substantially improved the overall flow and structure of the paper, ensuring that it falls into the category of a Research Article as opposed to a Case Study.

Experimental design

The introduction has been expanded and provides the necessary context for the research question, which is articulated at the end of the introduction. The methods are explained in a way so that they could be reproduced by others.

Validity of the findings

A conclusion section is now present, which addresses how the findings contribute to the current understanding of fee-based open access publishing. It would be interesting to also include questions for further study based on the stated conclusions.

Additional comments

No comments

Reviewer 2 ·

Basic reporting

No comments

Experimental design

No comments

Validity of the findings

No comments

Additional comments

The authors did a nice job addressing the concerns of the reviewers. I support acceptance of the paper for submission.

·

Basic reporting

The authors’ revisions have made a significant impact on the scope of this article. While the initial version presented an introduction and overview of the Open APC initiative, this paper is much more focused on Germany as a case study. The new title appropriately reflects these changes, as well as the research question. Overall, the original paper was an interesting look at Open APC, but this revision is a more practical exploration of the current state of affairs in open access publishing and will be useful for future studies.

Experimental design

The research question has been clearly defined and attempts to address the complexity of the fee-based open access publishing.

Validity of the findings

The discussion and conclusion sections provide a solid summary of the data and propose possible explanations for the findings.